# Reduced, Reused and Recycled: The Life of a Dataset in Machine Learning Research

**Bernard Koch**
University of California, Los Angeles
bernardkoch@ucla.edu

**Emily Denton**
Google Research, New York
dentone@google.com

**Alex Hanna**
Google Research, San Francisco
alexhanna@google.com

**Jacob G. Foster**
University of California, Los Angeles
foster@soc.ucla.edu

## Abstract

Benchmark datasets play a central role in the organization of machine learning research. They coordinate researchers around shared research problems and serve as a measure of progress towards shared goals. Despite the foundational role of benchmarking practices in this field, relatively little attention has been paid to the dynamics of benchmark dataset use and reuse, within or across machine learning subcommunities. In this paper, we dig into these dynamics. We study how dataset usage patterns differ across machine learning subcommunities and across time from 2015-2020. We find increasing concentration on fewer and fewer datasets within task communities, significant adoption of datasets from other tasks, and concentration across the field on datasets that have been introduced by researchers situated within a small number of elite institutions. Our results have implications for scientific evaluation, AI ethics, and equity/access within the field.

## 1 Introduction

Datasets form the backbone of machine learning research (MLR). They are deeply integrated into work practices of machine learning researchers, serving as resources for training and testing machine learning models. Datasets also play a central role in the organization of MLR as a scientific field. Benchmark datasets provide stable points of comparison and coordinate scientists around shared research problems. Improved performance on benchmarks is considered a key signal for collective progress. Such performance is thus an important form of scientific capital, sought after by individual researchers and used to evaluate and rank their contributions.

Datasets exemplify machine learning tasks, typically through a collection of input and output pairs [1]. When they institutionalize benchmark datasets, task communities implicitly endorse these data as meaningful abstractions of a task or problem domain. The institutionalization of benchmarks influences the behavior of both researchers and end-users [2]. Because advancement on established benchmarks is viewed as an indicator of progress, researchers are encouraged to make design choices that maximize performance on benchmarks, as this increases the legitimacy of their work. Institutionalization signals to industry adopters that models can be expected to perform in the real world as they do on the benchmark datasets. Close alignment of benchmark datasets with "real world" tasks is thus critical to accurate measurement of collective scientific progress and to safe, ethical, and effective deployment of models in the wild.

Given their central role in the social and scientific organization of MLR, benchmark datasets have also become a central object of critical inquiry in recent years [3]. Dataset audits have revealed concerning biases that have direct implications for algorithmic bias and harms [4, 5, 6, 7]. Problematic categorical

schemas have been identified in popular image datasets, including poorly-formulated categories and the inclusion of derogatory and offensive labels [8, 9]. Research into the disciplinary norms of dataset development has revealed troubling practices around dataset development and dissemination, like unstandardized documentation and maintenance practices [10, 11, 12]. There is also growing concern about the limitations of existing datasets and standard metrics for evaluating model behavior in real-world settings and assessing scientific progress in a problem domain [13, 14].

Despite the increase in critical attention to benchmark datasets, surprisingly little attention has been paid to patterns of dataset use and reuse across the field as a whole. In this paper, we dig into these dynamics. We study how dataset usage patterns differ across machine learning subcommunities and across time (from 2015-2020) in the Papers With Code (PWC) corpus.[1] More specifically, we study machine learning subcommunities that have formed around different machine learning tasks (e.g., *Sentiment Analysis* and *Facial Recognition*) and examine: (i) the extent to which research within task communities is concentrated or distributed across different benchmark datasets; (ii) patterns of dataset creation and adoption between different task communites; and (iii) the institutional origins of the most dominant datasets.

We find increasing concentration on fewer and fewer datasets within most task communities. Consistent with this finding, the majority of papers within most tasks use datasets that were originally created for *other* tasks, instead of ones explicitly created for their own task—even though most tasks have created more datasets than they have imported. Lastly, we find that these dominant datasets have been introduced by researchers at just a handful of elite institutions.

The remainder of this paper is organized as follows. First, we motivate our research questions by underscoring the critical importance of benchmarks in coordinating machine learning research. Second, we describe our analyses on the PWC corpus, a catalog of datasets and their usage jointly curated by the machine learning community (manually) and by Facebook AI Research (algorithmically). We then present our findings and discuss their implications for scientific validity, the ethical usage of MLR, and inequity within the field. We close by offering recommendations for possible reform efforts for the field.

## 2    Related Work: Scientific, Social, and Ethical Importance of Datasets

Following Schlangen [1], we understand machine learning benchmarks as community resources against which models are evaluated and compared. Benchmarks typically formalize a particular task through a dataset and an associated quantitative metric of evaluation. The practice was originally introduced to MLR after the "AI Winter" of the 1980s by government funders, who sought to more accurately assess the value received on grants [15, 16]. Today, benchmarking is the dominant paradigm for scientific evaluation in MLR, and the field collectively views upward trends on benchmarks as noisy but meaningful indicators of scientific progress [1, 2, 17]. Over time, MLR has evolved strong norms to facilitate widespread benchmarking, including the development of open-access datasets, formal competitions and challenges, and accompanying "black-box" software that allows researchers to test their algorithms on benchmark datasets with minimal effort.

The establishment of benchmark datasets as shared evaluative resources across the MLR community has unique advantages for coordinating scientists around common goals. First, barriers to participation in MLR are reduced, since well-resourced institutions can shoulder the costs of dataset curation and annotation.[2] Second, by reducing otherwise complex comparisons to a single agreed-upon measure, the scientific community can easily align on the value of research contributions and assess whether progress is being made on a particular task [19, 20]. Finally, a complete commitment to benchmarking has allowed MLR to relax reliance on slower institutions for evaluating progress like peer-review, qualitative or heuristic evaluation, or theoretical integration. Together, these advantages have contributed to MLR's unprecedented transformation into a "rapid discovery science" in the past decade [21].

While there are clear advantages to benchmarking as a methodology for comparing algorithms and measuring progress, there are growing concerns about benchmarking cultures in MLR that tend

---

[1]`https://paperswithcode.com`
[2]However, machine learning model development still remains a resource-intensive activity [18].

to valorize state-of-the-art (SOTA) results on established benchmark datasets over other forms of quantitative or qualitative analysis. The necessity of SOTA results on well-established benchmarks for publication has been identified as a barrier to the development of new ideas [22]. There have been growing calls for more rigorous and comprehensive empirical analysis of models beyond standard top-line metrics: reporting model size, energy consumption, fairness metrics, and more [23, 24, 25, 26]. The standard benchmarking paradigm also contributes to issues with underspecification in ML pipelines; a given level of performance on a held-out benchmark test set doesn't guarantee that a model has learned the appropriate causal structure of a problem [14]. In short, while community alignment on benchmarks and metrics can enable rapid algorithmic advancement, excessive focus on single metrics at the expense of more comprehensive forms of rigorous evaluation can lead the community astray and risk the development of models that generalize poorly to the real world.

The MLR community has begun to reflect on the utility of established benchmarks and their suitability for evaluative purposes. For example, the Fashion-MNIST dataset was introduced because the original MNIST dataset came to be perceived as over-utilized and too easy [27]; the utility of ImageNet — one of the most influential ML benchmarks in existence — as a meaningful measure of progress has been a focus of critical examination in the past few years [28, 29]. SOTA-chasing concerns are also compounded by the great capacity ML algorithms have to be "right for the wrong reason" [30], enabling SOTA results that rely on "shortcuts" rather than learning the causal structure dictated by the task [13]. Bender et al. suggest the NLP community may have been "led down the garden path" by over-focusing on "beating" benchmark tasks with models that can easily manipulate linguistic form without any real capacity for language understanding [31]. Recent dataset audits have also revealed that established benchmark datasets tend to reflect very narrow — typically white, male, Western — slices of the world [4, 5, 6, 7, 9]. Thus, over-concentration of research on a small number of datasets and metrics can distort perceptions of progress within the field and have serious ethical implications for communities impacted by deployed models. Despite these discussions, little empirical work has considered whether over-concentration of research on a small number of datasets is a systemic issue across MLR. This prompts our first research question:

**RQ1: How concentrated are machine learning task communities on specific datasets, and has this changed over time?**

There are also growing concerns regarding the gap between benchmark datasets and the problem domains in which they are used to evaluate progress. For example, Scheuerman et al. found that computer vision datasets tend to be developed in a manner that is decontextualized from a particular task or application area [12]. Supposedly "general purpose" benchmarks are often valued within the field, though the precise bounds of what makes a dataset suitable for general evaluative purposes remains unclear [17]. These observations prompt our second research question:

**RQ2: How frequently do machine learning researchers borrow datasets from other tasks instead of using ones created explicitly for that task?**

Despite widespread recognition that datasets are critical to the advancement of the field, careful dataset development is often undervalued and disincentivized, especially relative to algorithmic contributions [12, 32]. Given the high value the MLR community places on SOTA performance on established benchmarks, researchers are also incentivized to reuse recognizable benchmarks to legitimize their contributions. Dataset development is time- and labor-intensive, making large-scale dataset development potentially inaccessible to lower-resourced institutions. These observations prompt our final research question:

**RQ3: What institutions are responsible for the major ML benchmarks in circulation?**

Our paper makes two distinct contributions to the literature. First, it provides a concise, multi-dimensional discussion of the pros and cons of benchmarking as an evaluation paradigm in MLR, drawing on earlier work as well as insights from the sociology of science. Second, and more substantially, it provides the first field-level, quantitative analysis of benchmarking practice in MLR.

## 3   Data

Our primary data source is Papers With Code (PWC), an open source repository for machine learning papers, datasets, and evaluation tables created by researchers at Facebook AI Research. PWC is

largely community-contributed — anyone can add a benchmarking result or a task, provided the benchmarking result is publicly available in a pre-print repository, conference proceeding, or journal. Once tasks and datasets are introduced by humans, PWC scrapes arXiv using keyword searches to find other examples of the task or uses of the dataset.

We downloaded the complete PWC dataset on 06/16/2021 (licensed under CC BY-SA 4.0). In this study, we focus primarily on the "Datasets" archive, as well as papers utilizing those datasets. Each dataset in the archive is associated with metadata such as the modality of the dataset (e.g., texts, images, video, graphs), the date the dataset was introduced, and the paper title that introduced the dataset (if relevant). We found 4,384 datasets on the site and scraped 60,647 papers that PWC associates with those datasets using a PWC internal API (see Figure A2 for a truncated histogram of usage across datasets).

In PWC papers, benchmarks and datasets are associated with tasks (e.g., *Object Recognition*, *Machine Translation*). Because we are interested in the dynamics of dataset usage (both within and across task communities), our first two analyses are restricted to dataset usages published in papers annotated with tasks. We call the task for which the dataset was originally designed the "origin task." We call the task of the paper using the dataset the "destination task." For example, ImageNet [33] was introduced as a benchmark for *Object Recognition* and *Object Localization* (origin tasks), but is now regularly utilized as a benchmark for *Image Generation* (destination task) among many others.

PWC includes a taxonomy of tasks and subtasks. The graph is cyclic, making it hard to disentangle dataset transfer between broad tasks and finer-grained tasks. For each dataset transfer, we record the transfer between the origin task and the destination task. We also record the transfer between the origin's parents and the destination's parents. This approach allows us to accurately capture transfer dynamics between larger tasks (e.g., *Image Classification* and *Image Generation*), and between finer-grained tasks (e.g,. *Image-to-Image Translation* and *Image Inpainting*, which are both children of *Image Generation*).

We took three additional steps to pre-process the data. First, we only consider datasets that are used by others at least once. Second, because we found dataset usages in PWC to be noisy (i.e., a paper would be associated with a dataset if the corresponding dataset name appeared multiple times in the paper), we dropped dataset usages where the dataset-using paper shared no tasks in common with the dataset itself.[3] Second, we found 640 papers that introduced a dataset but were not associated with a task. Two authors manually annotated the top 90 most widely-used dataset papers with origin tasks (see GitHub for justifications and appendix for details). We dropped the remaining 550 dataset papers (accounting for only 10.2% of total usages).

**Datasets for Analysis 1 and 2 (RQ1, RQ2):** To minimize double-counting of dataset usages across parent tasks and child subtasks, we chose to focus exclusively on parent tasks in PWC. The outcome measures we use in these analyses (Gini, Adoption Proportion, and Creation Proportion) are biased in small samples, so we used only parent tasks above the median size of 34 papers (see GitHub for the list of tasks). Because these tasks were larger, we also felt that parent tasks tended to be more widely-recognized as coherent task communities. Table 1 presents descriptive statistics for the data used in each analysis. Analysis 1 explores dataset usage within tasks, so it includes datasets that are introduced in papers as well as those that are not (e.g., introduced on a website or competition). Analysis 2 explores the transfer of datasets between origin and destination tasks. This dataset is smaller because we can only determine the origin task for a dataset if it is introduced in a paper (Table 1). In the appendix, we describe robustness checks that remove some of these cleaning steps; these choices minimally affect our results.

**Dataset for Analysis 3 (RQ3):** To study the distribution of widely-utilized datasets across institutions, we linked all dataset-introducing papers to the Microsoft Academic Graph (MAG) [34]. Analyses were performed on dataset usages for which the dataset's last author affiliation was annotated in MAG (Table 1). We again imposed the restriction that usages must share a labeled task with the dataset, but again found it had minimal effects on the results (see appendix).

The datasets, a datasheet [10], and code for curation/analysis can be found at `https://github.com/kochbj/Reduced_Reused_Recycled`.

---

[3]Datasets in PWC are labeled with all tasks they are used for, not just the origin tasks. We focus on datasets introduced in papers so that we can identify tasks associated with the paper as origin tasks.

| Analysis | # Datasets | # Usages | # Tasks | # Papers |
|---|---|---|---|---|
| 1 | 2,063 | 49,008 | 133 | 26,691 |
| 2 | 960 | 33,034 | 133 | 20,747 |
| 3 | 1,933 | 43,140 | N/A | 26,535 |

Table 1: Descriptive statistics for data used in the three analyses. Note that the number of dataset usages is larger than the number of papers because many papers use multiple datasets.

## 4 Methods and Findings

### 4.1 Analysis 1 (RQ1): Concentration in Task Communities on Datasets

#### 4.1.1 Methods

To measure how concentrated task communities are on certain datasets (RQ1), we calculated the Gini coefficient of the distribution of observed dataset usages within each task. Gini is a continuous measure of dispersion in frequency distributions. It is frequently used in social science to study inequality [35].[4] The Gini score varies between 0 and 1, with 0 indicating that the papers within a task use all datasets in equal proportions, and 1 indicating that only a single dataset is used across all dataset-using papers. Gini is calculated as the average absolute difference in the usage of all pairs of datasets used in the task, divided by the average usage of datasets. Formally, if $x_i$ is the number of usages of dataset $i$ out of all $n$ datasets used in the task, then the Gini coefficient of dataset usage is,

$$G = \frac{\sum_{i=1}^{n}\sum_{j=1}^{n}|x_i - x_j|}{2\sum_{i=1}^{n}\sum_{j=1}^{n}x_j} = \frac{\sum_{i=1}^{n}\sum_{j=1}^{n}|x_i - x_j|}{2n\sum_{j=1}^{n}x_j} = \frac{\sum_{i=1}^{n}\sum_{j=1}^{n}|x_i - x_j|}{2n^2\bar{x}} \quad (1)$$

[5] Because Gini can be biased in small samples [36], we use the the sample-corrected Gini, $G_s = \frac{n}{n-1}G$, and excluded tasks (or task-years when disaggregating by time) with fewer than 10 papers.

**Regression Model 1:** In addition to descriptive statistics, we built a regression model to assess the extent to which observed trends in Gini from year-to-year could be attributable to confounding variables like task size, task age, or other task-specific traits at that time. Our outcome is $G_s$ in each task year from 2015-2020 (Figure A3 shows PWC coverage is limited for papers published before 2015). Our predictors of interest are:

1. **Year** (since we are interested in trends in concentration over time)
2. **CV, NLP, Methods**[6] (three dummy variables indicating whether the task belongs to the Computer Vision, Natural Language Processing, or Methodology categories in PWC).

To absorb additional variation, we also included the following control covariates:

1. **Task size** in number of dataset-using/introducing papers for that task in that year
2. **Task age** (because younger tasks may have higher Gini coefficients)
3. **Random intercepts for each task** (because we have repeated observations over time)

We use beta regression to model Gini because the beta distribution is very flexible, between 0 and 1, and commonly used for this purpose [35]. However, we apply the smoothing transformation in [37] to deal with the occasional task-year where the Gini is 0. We use a model with the following interactions:

$$\text{Beta}(G_s) = \alpha + \beta_1\text{Year} + \beta_2\text{TaskSize} + \beta_3\text{TaskAge}$$
$$\beta_4\text{CV} + \beta_5\text{NLP} + \beta_6\text{Methods} + \beta_7\text{Year} * \text{TaskSize} +$$
$$\beta_8\text{CV} * \text{Year} + \beta_9\text{NLP} * \text{Year} + \beta_{10}\text{Methods} * \text{Year}$$

This model was chosen among a set of nested models with two- and three-way interactions because it had the lowest Akaike information criterion (AIC) and Bayesian information criterion (BIC). See the appendix for model selection criteria and Table A1 for fit statistics.

---

[4]To give some indication of the range of Gini, the country with the lowest Gini for income inequality according to the World Bank [linked here] is Slovenia with a Gini of 24.6 (scaled 0 to 100). The country with the highest Gini inequality is South Africa at 63. The U.S. has a Gini of 41.4.

[5]Notation from Wikipedia, which provides an excellent exposition.

[6]Example "Methods" tasks in PWC include Transfer Learning, Domain Adaptation, and AutoML.

### 4.1.2 Findings

Controlling for task size, task age, and task-specific effects, Model 1 finds significant evidence for increasing concentration in task communities for the full dataset over time, predicting a marginal increase in Gini of 0.113 from 2015-2020 (Figure 1 top green; Table A2). This trend is also visible in the overall distributions of Gini coefficients over this period (Figure 1 bottom). By 2020, the median Gini coefficient for a task was 0.60. There are no statistically significant differences between Computer Vision and Methodology tasks compared to the full sample (Figure 1 top, Figure A1), but Model 1 suggests that increases in concentration are attenuated for Natural Language Processing task communities (Figure 1 top orange). We note that this is the only result that varies somewhat with our model specification; while the rate of increasing concentration in NLP tasks is always significantly lower than the rest of the dataset, the sign and slope of this change does vary somewhat across models. We discuss this point in the appendix.

## 4.2 Analysis 2 (RQ2): Changes in Rates of Adoption and Creation of Datasets Over Time

### 4.2.1 Methods

We created two proportions to better understand patterns of dataset usage and creation within tasks as outcomes:

$$\text{Adoption Proportion} = \frac{\text{\# of Papers Using Datasets from Other Tasks}}{\text{\# of Papers Using Datasets from Other Tasks} + \text{\# of Papers Using Datasets from this Task}}$$

$$\text{Creation Proportion} = \frac{\text{\# of Datasets Created Within this Task}}{\text{\# of Datasets Created within this Task} + \text{\# of Datasets Imported from Other Tasks}}$$

**Aggregated Descriptive Analyses:** We first computed these proportions for each of the 133 parent tasks aggregated across all years, and subsetted these by the "Computer Vision," "Natural Language Processing," and "Methodology" categories.

**Regression Models 2A & 2B:** Because we chose to formulate our outcomes as fractions of discrete events, logistic regression is the most theoretically appropriate model for these data. We used a mixed effects logistic regression to model these outcomes with the same predictors as Model 1.

### 4.2.2 Findings

The top row of Figure 2 shows a wide variance in adoption proportions in both the full sample and the subcategories. Within the full sample, more than half of task communities use adopted datasets at least 57.8% of the time. However, this number varies dramatically across the three PWC subcategories. In more than half of Computer Vision communities, authors adopt at least 71.9% of their datasets from a different task. The equivalent statistic in Methodology tasks is 74.1%. Conversely, half of Natural Language Processing communities adopt datasets less than 27.4% of the time.

In the bottom row of Figure 2, we see a largely inverted trend. Of all unique datasets used in a task community, 62.5% are created specifically for that task in more than half of tasks. Within Computer Vision and Methods tasks, the median is lower at 53.3% and 52.6%, with similar distributions across tasks. Most strikingly, 76.0% of datasets are created specifically for the task in more than half of NLP communities, with a much tighter variance.

We were unable to recover convincing evidence for trends in adoption or creation proportions either way (Regression Models 2A & 2B) because of a lack of data (results not shown). Disaggregating tasks over time creates a significant number of task-years with no events, and these metrics are undefined in those circumstances.

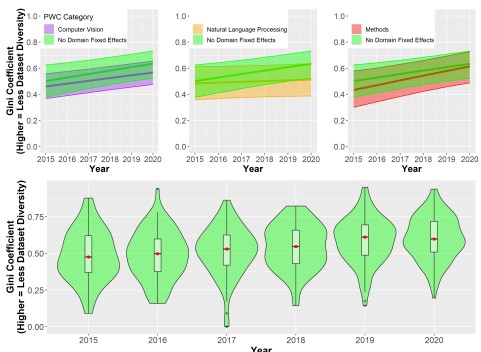

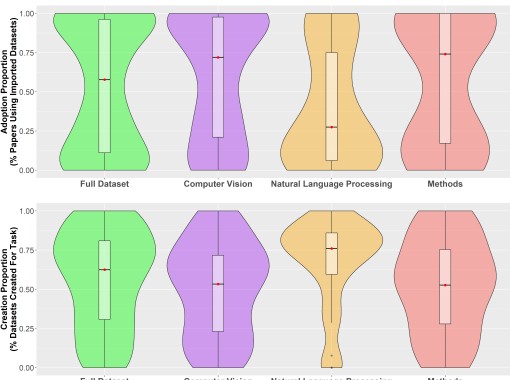

Figure 1: **Top: Predicted concentration on datasets across task communities over time.** Gini predicted by Model 1 holding task size/age at means. Green plots show the estimated effects of the full dataset, other colors are fixed effects for categories. 95% confidence intervals shown. **Bottom: Distributions of concentrations.** Higher Gini indicates greater concentration on fewer datasets. We observe significant spread of Gini across tasks, with the median increasing over time.

Figure 2: **Adoption (Top) and Creation (Bottom) Proportions for PWC Parent Tasks**. Full dataset in green, tasks in the Computer Vision category in purple, Natural Language Processing tasks in orange, and Methods tasks in red,. Red dot and line in boxplot indicate median. Width of violins indicates distribution of tasks.

## 4.3 Analysis 3 (RQ3): Concentration in Dataset-Introducing Institutions Over Time

### 4.3.1 Methods

To look at trends in Gini inequality across institutions and datasets over time for the larger set of dataset-using papers, we calculated the Gini coefficient $G_s$ in each year for dataset usages both by dataset and by institution. We regressed this Gini on year, as well as residuals capturing variance in the size of PWC that is not correlated with time (see appendix), using a standard beta regression. We also mapped dataset-introducing institutions using the longitude and latitude coordinates provided for the last author's institution on Microsoft Academic.

### 4.3.2 Findings

Overall, we find that widely-used datasets are introduced by only a handful of elite institutions (Figure 3 left). In fact, over 50% of dataset usages in PWC as of June 2021 can be attributed to just twelve institutions. Moreover, this concentration on elite institutions as measured through Gini has increased to over 0.80 in recent years (Figure 3 right red). This trend is also observed in Gini concentration on datasets in PWC more generally (Figure 3 right black).[7]

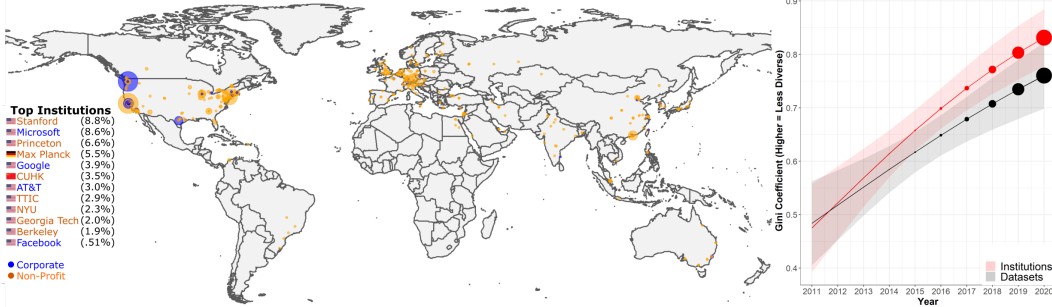

Figure 3: **Increases in concentration of dataset usages on institutions and datasets (non-task specific) over time. Left:** Map of dataset usages per institution as of June 2021. Dot size indicates number of usages. Blue dots indicates for-profit institutions and orange dots indicate not-for-profit. Institutions accounting for 50%+ of usages labeled. **Right:** Gini coefficient for concentration of dataset usages across the whole PWC dataset over time for both institutions and datasets. Ribbons indicate 95% CI; dot size indicates number of usages that year.

---

[7]The CIFAR-10/100 datesets were excluded from the institution analysis because the introducing dissertation is not in MAG, but are included in the dataset analysis. Inclusion would reduce majority-usage institutions to 7.

# 5 Discussion

In this paper, we find that task communities are heavily concentrated on a limited number of datasets, and that this concentration has been increasing over time (see Figure 1). Moreover, a significant portion of the datasets being used for benchmarking purposes within these communities were originally developed for a different task (see Figure 2). This result is striking given the fact that communities *are* creating new datasets — in most cases more than the unique number that have been imported from other tasks — but the newly created datasets are being used at lower rates. When examining PWC without disaggregating by task category, we find that there is increasing inequality in dataset usage globally, and that more than 50% of all dataset usages in our sample of 43,140 corresponded to datasets introduced by twelve elite, primarily Western, institutions.

NLP tasks differ somewhat from PWC as a whole: the broader trend of increasing concentration on a few datasets is moderated in NLP communities, new datasets are created at higher rates, and outside datasets are used at lower rates. One possible explanation for these findings is that NLP task communities in our dataset tend to be bigger than other task communities (median size of 76 dataset usages compared to 55). While we find very modest evidence of correlations between task size and adoption or creation proportions overall (Kendall's $\tau = -.008$, $p = .89$; $\tau = .014$, $p = .81$ respectively), these correlations are stronger within NLP tasks (Kendall's $\tau = -.10$, $p = .45$; $\tau = .09$, $p = .50$ respectively). It is possible that larger NLP communities are more coherent and thus generate and use their own datasets at higher rates than other task communities. Another possibility is that NLP datasets are easier to curate because the data are more accessible, easier to label, or smaller. The resolution of this puzzle is beyond the scope of this paper, but the distinct nature of NLP datasets provides an interesting direction for future work.

For our broader findings, there are valid reasons to expect widespread adoption and concentration on key datasets. First, a certain degree of research focus on a particular benchmark is both necessary and healthy to establish the validity and utility of the benchmark (or in some cases, to contest these properties) and to gain community alignment around the benchmark as a meaningful measure of progress. Second, the curation of large-scale datasets is not just costly in terms of resources, but may require unique or privileged data (e.g., anonymized medical records, self-driving car logs) accessible to only a few elite academic and corporate institutions. Nevertheless, the extent of concentration we observe poses questions relating to the scientific rigor and ecological validity of machine learning research and underscores benchmarking as a potential driver for inequality in the field. In the remainder of this section we discuss our findings in relation to these two broad themes and outline recommendations that can be enacted at an individual and institutional level. We close by discussing limitations of this analysis and outlining directions for future work.

## 5.1 Scientific Rigor and Ecological Validity of MLR

The heavy concentration of research on a small number of datasets for each task community is a fairly unsurprising result given the value placed on SOTA performance in established benchmark datasets —

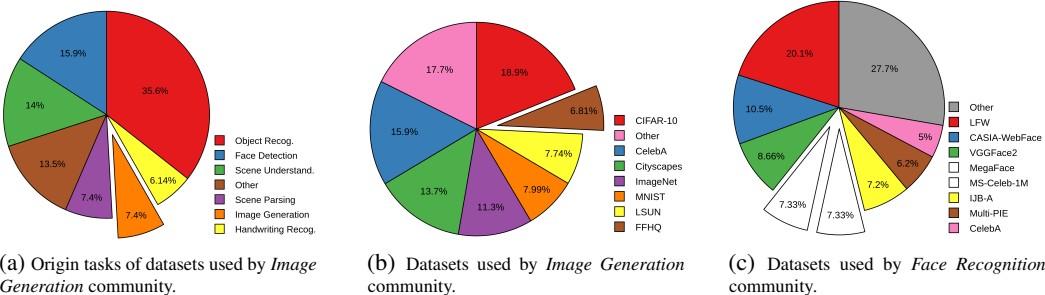

(a) Origin tasks of datasets used by *Image Generation* community.

(b) Datasets used by *Image Generation* community.

(c) Datasets used by *Face Recognition* community.

Figure 4: **Top datasets used across *Image Generation* and *Face Recognition* task communities**: (a) Origin task communities of top *Image Generation* datasets. Only 7.49% of *Image Generation* papers in PWC evaluate on datasets developed for *Image Generation*. (b) Names of top *Image Generation* datasets. Only one of the top datasets, FFHQ [38], was developed for the task. (c) The small number of datasets in usage within the high stakes domain of *Face Recognition*. Two of the datasets, MegaFace [39] and MS-Celeb-1M [40] (in white), have been recently retracted, the latter due to serious ethical violations [41].

a valuation that incentivizes individual researchers to concentrate on maximizing performance gains on well-established benchmarks. However, as discussed in Section 2, over-concentration of research efforts on established benchmark datasets risks distorting measures of progress. Moreover, as the rate of technology transfer has accelerated, benchmarks have been increasingly used by industry practitioners to assess the suitability and robustness of different algorithms for live deployment in production settings. This transition has transformed epistemic concerns about overfitting datasets into ethical ones. For example, critical research on datasets for facial recognition, analysis, and classification has repeatedly highlighted the lack of diversity in standard benchmark datasets used to evaluate progress [4], even as the technologies are applied in law enforcement contexts that adversely affect underrepresented populations [42]. Figure 4c shows the top datasets in usage within the *Face Recognition* community. Here, we see a significant amount of high stakes research being concentrated on a small number of datasets, many of which contain significant racial and gender biases [4, 43]. An in-depth examination of bias within the top benchmarks datasets in use within different task communities is outside the scope of this work. However, the systemic nature of bias concerns in ML datasets compounds the epistemic concerns associated with highly concentrated research.

Our findings also indicate that datasets regularly transfer between different task communities. On the most extreme end, the majority of the benchmark datasets in circulation for some task communities were created for other tasks. For example, Figure 4 plots the dataset usages of *Image Generation* papers on PWC broken down by origin task (Figure 4a) and dataset name (Figure 4b). We observe that only one of the datasets heavily used in the *Image Generation* community was designed specifically for this task. The widespread practice of adopting established datasets to train and evaluate models in new problem domains isn't inherently a problem. However, this practice does raise potential concerns regarding the extent to which datasets are appropriately aligned with a given problem space. Moreover, given the widespread prevalence of systematic biases in the most prominent ML datasets, adopting existing datasets, rather than investing in careful curation of new datasets, risks further entrenching existing biases.

Our findings on creation and adoption rates are quite nuanced. The extent to which high adoption rates raise significant concerns to ecological validity is yet to be determined. Furthermore, it is worth distinguishing between at least two forms of dataset adoption that seem to be conflated in the PWC data. On the one hand, we observe how datasets that have been developed for one task become *adapted* in some form for a new task through, for example, the addition of new annotations. On the other hand, we observe some datasets being *imported* whole cloth from one task community to another. Each of these forms of dataset adoption raises potentially unique concerns regarding the validity of the benchmark in a given context. That said, our results add empirical support to the growing body of scholarship calling for dataset development and use to be rooted in context [3, 12], which is particularly important for application-oriented tasks.

This paper complements and supports the growing calls to include forms of qualitative and quantitative evaluations beyond top-line benchmark metrics [23, 24, 25, 26]. Given the observed high concentration of research on a small number of benchmark datasets, we believe diversifying forms of evaluation is especially important to avoid overfitting to existing datasets and misrepresenting progress in the field.

### 5.2 Social Stratification in MLR

The extent of concentration we observe underscores that benchmarking is also a vehicle for inequality in science [44]. The *prima facie* scientific validity granted by SOTA benchmarking is generically confounded with the social credibility researchers obtain by showing they can compete on a widely recognized dataset, even if a more context-specific benchmark might be more technically appropriate. We posit that these dynamics creates a "Matthew Efffect" (i.e. "the rich get richer and the poor get poorer") where successful benchmarks, and the elite institutions that introduce them, gain outsized stature within the field [45].

Insofar as benchmarks shape the types of questions that get asked and the algorithms that get produced, current benchmarking practices offer a mechanism through which a small number of elite corporate, government, and academic institutions shape the research agenda and values of the field (Figure 3 left). Empirical support for this claim is beyond the scope of this paper, but there is work within the sociology of science and technology showing that government and corporate institutions tend to support research that serves (at least in part) their own interests, e.g., [46].

There is nothing *a priori* scientifically invalid about powerful institutions being interested in datasets or research agendas that would benefit them. However, issues arise when the values of these institutions are not aligned with those of other ML stakeholders (i.e., academics, civil society). For example, Dotan and Milli argue that deep learning's reliance on large datasets has forced MLR to confront decisions about the extent to which it is willing to violate privacy to acquire/curate data [2]. Corporate and government institutions have objectives that may come into conflict with privacy (e.g., surveillance), and their weighting of these priorities is likely to be different from those held by academics or AI's broader societal stakeholders. Returning to the Facial Recognition example in Figure 4c, four of the eight datasets (33.69% of total usages) were exclusively funded by corporations, the US military, or the Chinese government (MS-Celeb-1M, CASIA-Webface, IJB-A, VggFace2). MS-Celeb-1M was ultimately withdrawn because of controversy surrounding the value of privacy for different stakeholders [41].

The recently introduced NeurIPS Dataset and Benchmark Track is a clear example of an intervention that shifts incentive structures within the MLR community by rewarding dataset development and other forms of data work. We believe these sorts of interventions can play a critical role in incentivizing careful dataset development that is meaningfully aligned with problem domains. However, our finding that a small number of well-resourced institutions are responsible for most benchmarks in circulation today has implications for data-oriented interventions in the field. Our research suggests that simply calling for ML researchers to develop more datasets, and shifting incentive structures so that dataset development is valued and rewarded, may not be enough to diversify dataset usage and the perspectives that are ultimately shaping and setting MLR research agendas. In addition to incentivizing dataset development, we advocate for equity-oriented policy interventions that prioritize significant funding for people in less-resourced institutions to create high-quality datasets. This would diversify — from a social and cultural perspective — the benchmark datasets being used to evaluate modern ML methods.

### 5.3  Limitations and Future Work

Because our findings rely on a unique community-curated resource, our results are contingent on the structure and coverage of PWC. Sensitivity analyses suggest that while PWC's coverage of ML publications is not perfect and exhibits some recency bias, the omitted papers tend to be low impact. Moreover, the crowdsourced taxonomy of parent-child task relations in PWC may be subjective and/or noisy, especially for small or new tasks.[8] To increase our confidence in task annotations, we focused our analyses on larger, higher-level task communities and considered dataset usages invalid if they did not share a task label with the dataset. Lastly, we find that the concentration trends in Regression 1 are largely robust to model specification and our choice of Gini as an outcome. See the appendix for details on design choices and sensitivity analyses.

Finally, we emphasize that our findings are highly nuanced. We report trends that our analyses revealed, but refrain from imposing normative judgements on many of these trends. For example, the high rates of adoption raise potential concerns and point to an important future area of examination. The mere fact that datasets travel between task communities is not necessarily problematic, and indeed the widespread sharing of datasets has been central to methodological advancements in the field. We hope this work will offer a foundation for future empirical work examining the details of dataset transfer and the context-specific implications of our findings.

## 6  Conclusion

Benchmark datasets play a powerful role in the social organization of the field of machine learning. In this work, we empirically examine patterns of creation, adoption, and usage within and across MLR task communities. We find that benchmarking practices are heavily concentrated on a small number of datasets for each task community and heavily concentrated on datasets originating from a small number of well-resourced institutions across the field as a whole. We also find that many benchmark datasets flow between multiple task communities and are leveraged to evaluate progress on tasks for which the data was not explicitly designed. We hope this analysis will inform community-wide initiatives to shift patterns of dataset development and use so as to enable more rigorous, ethical, and socially informed research.

---

[8]The full list of parent tasks and parent/child relations is available in the GitHub.

## Acknowledgments and Disclosure of Funding

We thank the reviewers for their helpful comments, and Siavash Jalal for statistical advice. The authors have no competing interests to disclose. BK was supported by DDRIG and GRFP grants from the US National Science Foundation. JGF was supported by an Infosys Membership in the School of Social Science at the Institute for Advanced Study.

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
