# Reduced, Reused and Recycled: The Life of a Dataset in Machine Learning Research

## A Appendix

### A.1 Sensitivity Analyses

We conducted three types of sensitivity analyses to gauge the extent of biases stemming from the Papers with Code corpus, our own curatorial decisions, and model selection/robustness. We describe each of these analyses below, as well as potential limitations still unaddressed.

**Coverage biases in PWC**

The validity of our findings is contingent on PWC representing the field of MLR more broadly. To estimate the extent of coverage bias and recency bias in PWC, we searched PWC for all papers published in ten top ML conferences (NeurIPS, ICML, ICLR, ACL, AAAI, AISTATS, KDD, CVPR, SIGIR, IJCAI) between 2015 and 2020 (46,774 papers) according to Microsoft Academic. We found that 58.9% of these papers appeared in PWC. However these 58.9% of papers accounted for 89.3% of collective citations received by the 46,774 papers, suggesting that the missing papers are primarily lower impact papers. When we disaggregated this analysis by year, we find that coverage ranges from 38.9% of 2015 papers to 68.8% of 2020 papers, but the proportion of citations covered in each year from 2015 on never drops below 86%. While these numbers suggest that omitted papers may be less cited and thus unlikely to propose widely used datasets, we note that they do not address possible under-annotation of dataset creations or usages within included PWC papers.

**Robustness to Cleaning Decisions**

*Focus on larger, higher-level tasks*

Another set of possible biases has to do with subjective, unreliable, and/or spurious task annotations in PWC. To our understanding, the task ontology in PWC is purely crowd-sourced. We chose to focus on higher-level tasks in PWC that were parents to others because we wanted to minimize the double counting of dataset transfers, and were concerned that some of the smaller tasks might represent idiosyncratic labels by individuals rather than stable problem communities. We also chose to focus on the top 50% of parent communities with the most dataset usages (133 task communities with more than 34 usages, the median number of usages). This choice allowed us to reduce sample-size-related issues in our regression analyses and further sharpened our focus on the largest, most popular tasks. Below we describe how these choices affect Analyses 1 and 2 (Analysis 3 is task-independent).

For Analysis 1, when we focus on all 269 parent tasks instead of the 133 largest, there is still a significant increase in Gini over time for all but the smallest tasks. However, the interaction with NLP becomes statistically insignificant. When we do not disaggregate by time, the median Gini for the whole dataset drops from 0.72 to 0.65 (still quite large), with the smaller task communities tending to have lower Gini coefficients. If we include both children and parent tasks, this creates a different dataset by adding 1,164 additional tasks with a median task size of 7 aggregated across all years, and 3 when disaggregated across years. The significant increase in concentration over time is preserved, as is the significant interaction with NLP. Median Gini drops to 0.48 in this dataset. Nevertheless, we discourage interpretation of these findings because Gini is known to be biased in small samples [36].

For Analysis 2, when we focus on all 269 parent tasks instead of the 133 largest, the median creation proportion is stable at 62.5%. The median adoption proportion increases from 57.7% to 69.7%. Again we see similar patterns for the NLP community, where the median creation proportion is higher (75%) than the full dataset and the median adoption proportion is lower at 40%. When we include transfers between all 1,025 tasks, the proportions become increasingly biased as the median task size dips to 7, with a median creation proportion of 33% and a median adoption proportion of 100% (because community sizes are so small).

*Cleaning of spurious annotations*

Our analyses on the cross-task transfer of datasets required that the dataset-introducing paper be labeled with origin tasks and the dataset-using papers be labeled with destination tasks. However,

we found that some of the most widely-used datasets in PWC had no task annotations (e.g., MNIST, CIFAR-10, CelebA, ImagenNet). To include these datasets in our analyses, two authors went through each dataset-introducing paper and extracted evidence for specific PWC task labels. These annotations and justifications can be found in the GitHub. Starting with the raw data, we scraped 92,874 dataset usages from 46,697 dataset-using PWC papers labeled with tasks. Only 49,589 (53%) of those were to datasets already labeled with tasks in PWC. We first labeled the 45 highest-used datasets with their original tasks, skipping datasets that did not seem designed for MLR or where origin tasks were unidentifiable from language used in the paper or website. By manually labeling the 90 largest datasets with tasks, we recovered 33,739 usages, leaving just 10.2% of total dataset usages unlabeled with tasks across 550 datasets. It was too time-intensive for us to label these last 550 datasets. However, our results do not change when we include only 45 manually annotated datasets versus 90.

We can also relax the requirement that both the dataset and the focal dataset-using paper be labeled with an overlapping task in Analysis 3. For this analysis, this relaxation allows us to consider 78,289 usages (primarily algorithmically-labeled) of 2,174 datasets in 46,842 papers. Including these potentially noisy usages does not affect our findings: Gini still increases from 0.31 to 0.86 from 2011-2020 across datasets and from 0.38 to 0.80 across institutions. The number of institutions accounting for 50%+ of usages shrinks to 9 (from most usages to least): Princeton, Stanford, Microsoft, AT&T, Max Planck, CUHK, Google, NYU, Toyota Technical Institute at Chicago.

**Robustness to Model Design Choices**

*Using Entropy-based metric instead of Gini*

While we ultimately chose to use Gini inequality as the outcome for Model 1, there are alternative metrics for the evenness of categorical distributions. In addition to Gini, we performed all concentration analyses using the information-theoretic Pielou evenness [47] $J(y)$. Pielou evenness normalizes the observed Shannon entropy $H(y)$ of dataset usages in each task-year by the maximum possible entropy in each task-year (i.e., the scenario where all datasets are used equally):

$$J(y) = \frac{H(y)}{H'_{\max}(y) = -\sum_{i=1}^{D} \frac{1}{D} \ln \frac{1}{D} = \ln D} \tag{2}$$

where $D$ is the number of datasets used at least once in the task-year.

Like Gini, Pielou evenness is between 0 and 1, but interpretation runs in the opposite direction to Gini: higher numbers indicate high evenness (close to maximum entropy/maximum evenness), and low numbers indicate high concentration.

At 0.69, the median Pielou evenness across parent tasks is fairly high. However it behaves identically to Gini in regression analyses; the same model specification is selected, the same parameters are significant (Table A3), and the same trends are observed (Fig. A4).

*Model Selection for Regression 1*

The last possible source of bias that we consider in our findings comes from the specification of our models. We started with a fully restricted model presented below:

$$\text{Beta}(G_s) = \alpha + \beta_1 \text{Year} + \beta_2 \text{TaskSize} + \beta_3 \text{TaskAge}$$
$$\beta_4 \text{CV} + \beta_5 \text{NLP} + \beta_6 \text{Methods} + \beta_7 \text{Year} * \text{TaskSize} + \beta_8 \text{Year} * \text{TaskAge}$$
$$\beta_9 \text{CV} * \text{Year} + \beta_{10} \text{NLP} * \text{Year} + \beta_{11} \text{Methods} * \text{Year} +$$
$$\beta_{12} \text{Year} * \text{TaskAge} * \text{TaskSize}$$

We then compared various simpler models where we dropped three-way interactions, two-way interactions, and random intercepts to this fully restricted model using the Akaike information criterion (AIC) and Bayesian information criterion (BIC). Model fit metrics are provided in Table A1. Exponentiated coefficients for the best-performing model (presented in the main text) under both AIC and BIC are presented in Table A2. In general we found the coefficients are largely stable across the various models.

*Model Specification for Regression in Analysis 3*

Because we were concerned that the growth in PWC over time might have been a confounder for changes in Gini across institutions or datasets, we included residuals for the size of PWC after it was exponentially regressed on time as a covariate. Ultimately this coefficient was insignificant and did not affect our results.

## A.2 Supplemental Tables

Table A1: Fit statistics for Regression 1. AIC is the Aikeike criterion. BIC is the Bayesian Information Criterion. ICC is the intraclass correlation coefficient for the different task communities.

| Model | AIC | BIC | ICC |
|---|---|---|---|
| Fully Restricted | -725.050 | -656.208 | **1.027** |
| No 3-Way Interaction | -724.400 | -664.163 | 1.026 |
| No *Year*Task Age* | **-725.108** | **-669.174** | **1.027** |
| No *Year*Task Size* | -696.567 | -640.633 | 1.025 |
| No *Year*Task Age*, *Year*Task Size*, or Random Intercepts | -478.350 | -431.021 | N/A |

Table A2: Exponentiated coefficients for fixed effects in Regression Model 1. Colons indicate interactions. Bolding highlight coefficients with $p < 0.05$

| Term | Exponentiated $\beta$ | Std Error | Z-Value | P-value |
|---|---|---|---|---|
| Intercept | 1.21 | 1.22 | 0.95 | 0.34 |
| Year | 1.10 | 1.04 | 2.31 | **0.02** |
| Task Size | 2.39 | 1.15 | 6.14 | **0.00** |
| Task Age | 0.94 | 1.06 | -1.06 | 0.29 |
| CV | 0.87 | 1.21 | -0.72 | 0.47 |
| NLP | 1.02 | 1.23 | 0.10 | 0.92 |
| Methodology | 0.74 | 1.21 | -1.6 | 0.11 |
| Year:Task Size | 0.86 | 1.03 | -5.59 | **0.00** |
| Year:CV | 0.98 | 1.04 | -0.63 | 0.53 |
| Year:NLP | 0.92 | 1.04 | -2.17 | **0.03** |
| Year:Methodology | 1.04 | 1.04 | 0.99 | 0.32 |
| SD(Task Random Intercepts) | 1.71 | | | |

Table A3: Exponentiated coefficients for fixed effects in Regression Model 1 using Pielou evenness instead of Gini. Colons indicate interactions. Bolding highlight coefficients with $p < 0.05$

| Term | Exponentiated $\beta$ | Std Error | Z-Value | P-value |
|---|---|---|---|---|
| Intercept | 3.53 | 1.26 | 250 | **0.00** |
| Year | 0.89 | 1.04 | 0.01 | **0.01** |
| Task Size | .523 | 1.14 | 0.00 | **0.00** |
| Task Age | 1.06 | 1.07 | 2.2 | 0.43 |
| CV | 1.31 | 1.25 | 3.35 | 0.23 |
| NLP | 1.04 | 1.28 | 1.19 | 0.85 |
| Methodology | 1.57 | 7.04 | -1.6 | 0.05 |
| Year:Task Size | 1.13 | 107 | -5.59 | **0.00** |
| Year:CV | 1.03 | 1.04 | 2.21 | 0.42 |
| Year:NLP | 1.12 | 1.04 | 14.2 | **0.01** |
| Year:Methodology | 0.93 | 1.04 | .19 | 0.09 |
| SD(Task Random Intercepts) | 2.00 | | | |

## A.3 Supplemental Figures

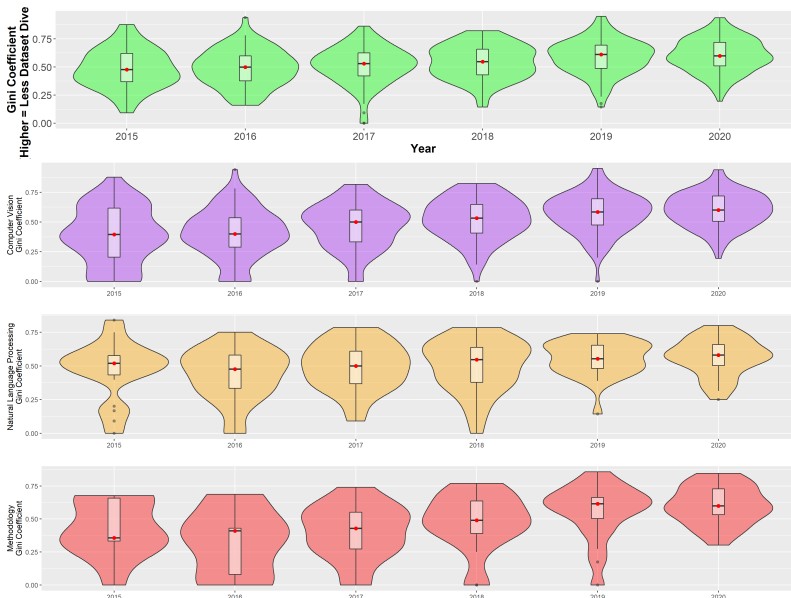

Figure A1: **Increases in concentration on datasets within task communities over time.** Higher Gini coefficient indicates greater concentration on fewer datasets. We observe significant spread of Gini across different task communities, with the median trending upwards over time for all modalities. Green is the full dataset, other colors indicate subsets of the data by PWC task category.

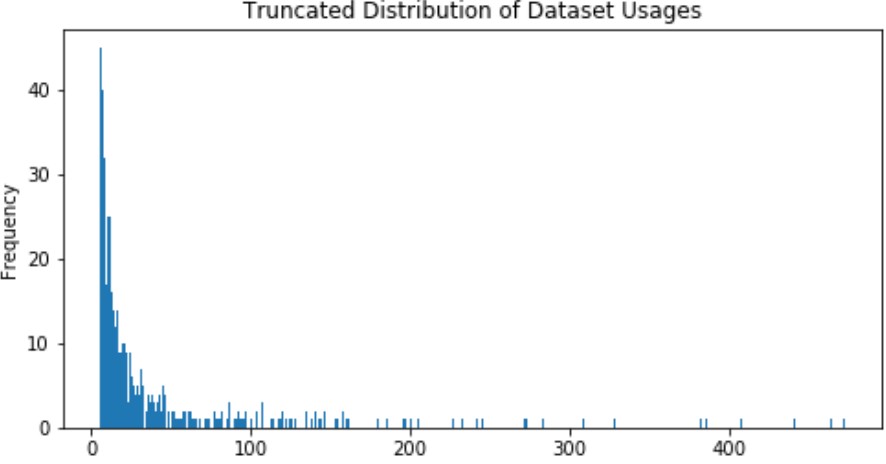

Figure A2: **Truncated distribution of usages per dataset in PWC.** Usages measured conservatively by only allowing usages from tasks the dataset was labeled for. 3760 datasets with less than 5 papers and 8 dataset with over 500 uses dropped for clarity. These 8 dropped datasets are: Penn Treebank, CelebA, SQuAD, KITTI, MNIST, Cityscapes, ImageNet, COCO.

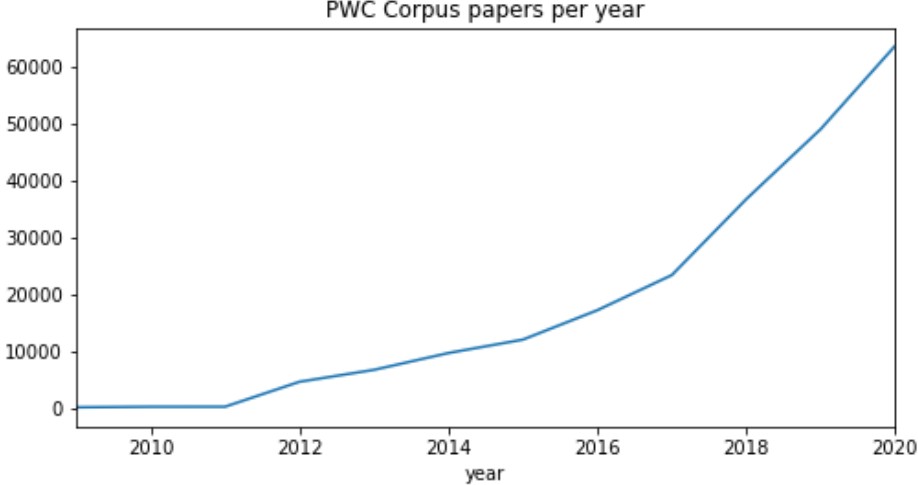

Figure A3: **Number of Papers in the Papers with Code Corpus**. Full set of "Papers with Abstracts" on Papers with Code as of June 2021. Total dataset size is 137,510 papers. Daily snapshots of this dataset are available at github.com/paperswithcode.

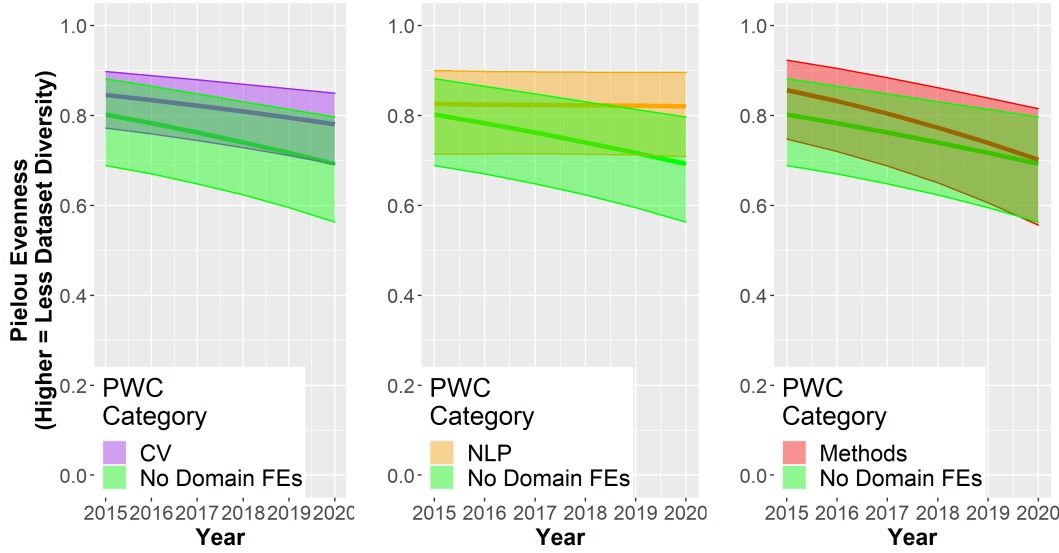

Figure A4: **Predicted Pielou evenness on datasets across task communities over time.** Pielou evenness predicted with same specification as Model 1 holding task size/age to means. Green plots show the estimated effects of the full dataset, other colors are fixed effects for categories. 95% confidence intervals shown.