# OpenReview forum: "Reduced, Reused and Recycled: The Life of a  Dataset in Machine Learning Research"
_NeurIPS.cc/2021/Track/Datasets_and_Benchmarks/Round2 — NeurIPS 2021 Datasets and Benchmarks Track (Round 2)_

### Official Review · Reviewer_WdbG · 2021-09-14
**A Concise Delineation of Trends in Machine Learning Benchmarking**

**Rating:** 8
**Confidence:** 4
**Clarity:** The paper is crystal clear, and beaut…

**Strengths:**

Overall: The paper is very concisely written, and extremely easy to follow.

No small feat: the authors identify over 2.5k unique datasets from over 45k entries parsed from Papers With Code.

Analysis: thorough analysis common trends in MLR benchmarking, and the adoption and creation of datasets.

Relevance: this paper is extremely relevant to anybody involved in machine learning research. Simple in nature, yet detailed enough to convince readers of their findings. I can very clearly picture this paper being an assigned reading at university courses.



**Weaknesses:**

Related work: Are there other papers that study dataset & benchmark trends? If so, what are their findings? It may be helpful to include these in a related work section. If no such papers exist, simply mentioning that this is the case may further motivate your paper.

Beta regression: my understanding is that the regression model 1 was selected amongst all models with varying interaction terms. It may be helpful to to add the resulting AIC criterion and potentially other goodness-of-fit statistics to the appendix to give a bit more context about the selected model. This is however a relatively nominal concern.

**Additional Feedback:**

Overall, the paper is interesting, very well written and delivers a concise punchline. However, my main concern is that without a "related work section" it seems too removed from prior work in the benchmarking community.


in addition to all your discoveries, I believe that the paper is a great opportunity to showcase your new "dataset of datasets" sourced from PWC. The crawled data can be emphasized throughout the paper not just as a pre-processing step, but rather as one of your main contributions to the community.



**Correctness:**

The claims made in the submission are sound, and empirically demonstrate overall trends present in the literature. The statistical methods used in the paper are justified, although for completeness I recommend adding adding more details about the regression model 1 in the appendix.

**Documentation:**

The data-collection process, namely the retrieval and parsing of the datasets & papers from Papers With Code (PWC), is well documented in the paper. However, more specific instructions for the usage of the dataset sourced from PWC could be added to the main paper.

**Ethics:**

The paper makes interesting discoveries regarding the inequality of MLR benchmarks. However, in section 5.2, there is a claim that feels unsubstantiated.

The paper soundly concludes that a number of well-resourced institutions are responsible for most benchmarks. However, there is no evidence presented that leads me to believe that elite institutions are able to push their agenda by dominating the benchmarking scene. For instance, how can we ascertain the causality of institutions influencing benchmarks, which in turn influence research directions of sub-communities?

Even though I am very much inclined to believe that the aforementioned hypothesis is true, I believe it would be best to present it as future work rather than a conclusion.

**Relation To Prior Work:**

This area is not specified in the paper. Mentioning related work (or lack of thereof) can bolster the overall paper.

**Summary And Contributions:**

The authors study current trends in benchmarking and datasets in machine learning research (MLR).

By analyzing thousands of papers sourced from Papers With Code (PWC), they study the evolution of dataset concentration across communities over time, and the interplay between dataset adoption and creation for varying sub-communities (e.g. computer vision, NLP and methods). Their results show that most communities display an upward trend in concentration, effectively focusing focusing more on fewer datasets.

Further, the authors look into the institutional concentration of dataset creation, and show that the majority of common MLR benchmarks originate from "a handful of elite institutions". They argue that this inequality can be detrimental to overall progress in MLR.

---

> ### Author Response · Authors · 2021-09-30
> **Response**
>
> Thank you very much for the interesting feedback, particularly about the power of institutions.
>
> **Related work: Are there other papers that study dataset & benchmark trends? If so, what are their findings? It may be helpful to include these in a related work section. If no such papers exist, simply mentioning that this is the case may further motivate your paper.**
>
> Section 2 was originally titled “Related Work”, but we decided it would be clearer to structure it more like the literature review of a social science paper that motivates research questions. We agree that the narrative style may be unfamiliar to ML researchers, so we have prepended “Related Work” to the section header.
>
> While we are not aware of other field-scale quantitative studies of benchmarking, we have added a few more sources with excellent meta-level discussions of benchmarking from within computer science:
> - Susan Sim’s ICSE paper describing how benchmarking communities resemble Kuhnian paradigms.
> - Ken Church’s discussions of the historical origins of benchmarking within MLR.
> For the interested reader, we note that there were several excellent workshops discussing benchmarking practice at [NIST](https://www.nist.gov/news-events/events/2021/06/ai-measurement-and-evaluation-workshop) and [ACL](https://github.com/kwchurch/Benchmarking_past_present_future/blob/master/README.md#Makhoul)  in 2021.
>
> If there are other papers or research areas which the reviewers feel we have overlooked, we would appreciate recommendations!
>
> Lastly, we have added two sentences at the end of Section 2 that clarify the unique position of this paper within the relevant literature:
>
> “Our paper makes two distinct contributions to the existing literature. First, it provides a concise, multi-dimensional discussion of the pros and cons of benchmarking as an evaluation paradigm in MLR, drawing on earlier work as well as insights from the sociology of science. Second, and more substantially, it provides the first field-level, quantitative analysis of benchmarking practice in MLR to our knowledge.”
>
> **Beta regression: my understanding is that the regression model 1 was selected amongst all models with varying interaction terms. It may be helpful to to add the resulting AIC criterion and potentially other goodness-of-fit statistics to the appendix to give a bit more context about the selected model. This is however a relatively nominal concern.**
>
> We will add these details to the appendix. For a more detailed response to some of these modeling choices, please see our response to Reviewer #2.
>
> **In addition to all your discoveries, I believe that the paper is a great opportunity to showcase your new "dataset of datasets" sourced from PWC. The crawled data can be emphasized throughout the paper not just as a pre-processing step, but rather as one of your main contributions to the community.**
>
> We appreciate the reviewer’s acknowledgment of the effort that went into curating and cleaning this “dataset of datasets.” In the submitted draft, we did not emphasize the cleaned “dataset of datasets” as a contribution for several reasons: limited space, the fact that we think this data will age quickly as the field evolves, and the fact that the PWC team has a done an excellent job already of releasing parsable data. From our perspective, the additional curation steps we contributed are:
> - Reconstructing the task ontology from benchmarks in PWC. Since we began the project, PWC has since made the ontology editable directly by users, so it is possible that relations based only on benchmarks do not capture all links.
>  - Identifying dataset-citing papers using an undocumented but already implemented API. Given the PWC team’s commitment to transparency, these features may become publicly available at some point in the future if there is demand.
> - Collating the multiple paper datasets in PWC which are not totally overlapping (i.e., papers with benchmarks, papers with datasets, papers with abstracts, and papers that use datasets).
> - Cleaning dataset usages to be more conservative than just keyword highlighting.
> - Manually annotating major dataset-introducing papers with their original tasks in PWC.
> - Linking paper titles to Microsoft Academic Graph IDs.
>
> All this being said, we pledge to release code that will show how to construct the dataset and a complementary datasheet.

---

> > ### Author Response · Authors · 2021-09-30
> > **Response continued**
> >
> >
> > **The paper makes interesting discoveries regarding the inequality of MLR benchmarks. However, in section 5.2, there is a claim that feels unsubstantiated. The paper soundly concludes that a number of well-resourced institutions are responsible for most benchmarks. However, there is no evidence presented that leads me to believe that elite institutions are able to push their agenda by dominating the benchmarking scene. For instance, how can we ascertain the causality of institutions influencing benchmarks, which in turn influence research directions of sub-communities.**
> >
> > **Even though I am very much inclined to believe that the aforementioned hypothesis is true, I believe it would be best to present it as future work rather than a conclusion.**
> >
> >    We thank the reviewer for raising this point and welcome dialogue in the spirit of this open review process. Our original claim (in the “Discussion” section) was: “To the extent that benchmarks shape the types of questions that get asked and algorithms that get produced, current benchmarking practices thus offer a mechanism through which a small number of elite institutions, both academic and for-profit, can shape the agenda of the field.”
> >
> > It is true that we do not provide evidence for this deduction from our empirical finding that elite institutions dominate the production of widely-used benchmarks, and we have softened the wording. However, we think it follows from reasonable statements:
> >
> > 1. The benchmarking process and datasets used shape the types of questions that get asked and algorithms that get produced within the field.
> > 2. Current practices involve coordinating research on a small number of datasets and those datasets tend to be produced by elite institutions.
> > 3. Elite government and corporate institutions may tend to produce datasets that reflect their interests out of the larger space of possibilities people may be interested in. We have no evidence for this, but we think it is a reasonable deduction. There are plenty of works in the sociology of science and technology which argue that powerful institutions tend to support research that serves (at least in part) their own interests (Oreskes 2021). As Oreskes argues, there is nothing a priori scientifically invalid about powerful institutions being interested in some questions rather than others, or in the existence of some technologies (which would benefit them) rather than others (which might not).
> >
> > In MLR, issues arise when the values of government and corporate institutions come into conflict with the values of other AI stakeholders (e.g., academics, broader society). For example, Dotan and Milli (2019) argue that deep learning’s reliance on large datasets has forced MLR to confront decisions about the extent to which it is willing to violate privacy to acquire/curate data. Corporate and government institutions have objectives that may come into conflict with privacy (e.g., surveillance), and their weighting of these priorities is likely to be different from those held by academics or AI’s broader societal stakeholders. Returning to the Facial Recognition example in Figure 4c, four of the seven datasets (41.2% of total usages) were exclusively funded by corporations, the US military, or the Chinese government (MS-Celeb 1M, CASIA-Webface, IJB-A, VggFace2). MS-Celeb 1M was ultimately withdrawn because of controversy surrounding the value of privacy for different stakeholders.
> >
> > Dotan, Ravit, and Smitha Milli. "Value-laden disciplinary shifts in machine learning." arXiv preprint arXiv:1912.01172 (2019).
> >
> > Oreskes, Naomi. Science on a Mission: How Military Funding Shaped What We Do and Don’t Know about the Ocean. United States: University of Chicago Press, 2021.

---

> > > ### Comment · Reviewer_WdbG · 2021-09-30
> > > **Reply to authors' response**
> > >
> > > Thank you for thoroughly addressing my questions and concerns. In light of your responses, I have updated my score to 8: Top 50%

---

### Official Review · Reviewer_4yHW · 2021-09-19
**What PWC can tell us about Machine Learning Research**

**Rating:** 8
**Confidence:** 4
**Clarity:** The paper is well-written and easy to…

**Strengths:**

1. Even though the results are not surprising, these analyses were necessary to validate presumptions one might have with scientific rigor. It is an important topic that was not previously addressed by the ML community in publication.
2. The description of work motivation, dive into the importance of benchmark datasets, and discussion section are well-written and valuable on their own due to providing meta-level remarks on the scientific process in ML in general.

**Weaknesses:**

1. As the authors are perfectly aware, some weaknesses result from the choice of PWC as a material. Factors such as the arbitrary nature of assigned labels and recency bias could affect obtained results. Ideally, I would see yet another analysis of how representative PWC is for the ML community. For example, some of the mentioned biases' prevalence can be estimated by comparison to a small portion of ML papers sampled from a different source. Others can be considered by assessing the level of noise in PWC annotations. I am aware of how time- and space-consuming would be the fulfillment of this suggestion, but maybe you can elaborate a little bit on the topic in a quantitative manner.
2. I believe some details not covered in the Appendix should be placed here, e.g., the calculated β estimates, more information on conducted AIC-based model selection, and a description of the reasoning that led you to choose linear or logistic regression in a particular scenario.

**Additional Feedback:**

I noticed several minor problems:
- you write that $p = 0$ in #234, I believe it is a typo and there should be some non-zero value,
- "e.g." in #46 lacks a comma despite the fact you add it in other parts of the paper,
- footnote in #73 should be placed after the period,
- "datset" instead of "dataset" in #169.



**Correctness:**

Though parts of the process are debatable (e.g., due to insufficient description), I believe the analyses were conducted in a rather sound way and are correct.

**Documentation:**

This criterium largely does not apply due to the meta- and discussion nature of the paper. One can reproduce the obtained results with ease given the provided details.


**Ethics:**

There are no ethical concerns to rise.

**Relation To Prior Work:**

Relation to prior work is clearly discussed in the introduction section.

**Summary And Contributions:**

**Update.** Thank you for addressing my concerns and providing detailed comments.
___

The paper describes analyses conducted on the Paper With Codes dataset to diagnose:

1. The concentration of the ML community w.r.t. tackled datasets (in terms of statistical dispersion over tasks).
2. How common is a trend of adapting existing datasets to new ML problems?
3. What are the affiliations of the most popular benchmark authors?

It is concluded that the said dataset concentration increased over the years 2015-2020, the trend of adapting existing datasets is common, and the most popular benchmarks were proposed by authors affiliated with several elite institutions.

---

> ### Author Response · Authors · 2021-09-30
> **Response**
>
> We appreciate this reviewer’s overall positive assessment of our contributions and are grateful for the places where they suggested improvement. Addressing these sensitivity and transparency concerns has strengthened the manuscript considerably.
>
> **Weakness #1: As the authors are perfectly aware, some weaknesses result from the choice of PWC as a material. Factors such as the arbitrary nature of assigned labels and recency bias could affect obtained results. Ideally, I would see yet another analysis of how representative PWC is for the ML community. For example, some of the mentioned biases' prevalence can be estimated by comparison to a small portion of ML papers sampled from a different source. Others can be considered by assessing the level of noise in PWC annotations. I am aware of how time- and space-consuming would be the fulfillment of this suggestion, but maybe you can elaborate a little bit on the topic in a quantitative manner.**
>
> We appreciate your constructive comments on dataset bias. Below we list all possible sources of bias we can think of and the extent to which we can address them. We will include this discussion in a datasheet accompanying the final paper, and have added remarks about the sensitivity analyses described below to the  "Limitations and Future Work" section.
>
> - There is coverage bias stemming from the fact that PWC is not a representative sample of the field of MLR. To assess coverage bias we downloaded all papers published in ten top ML conferences from MAG (NeurIPS, ICML, ICLR, ACL, AAAI, AISTATS, KDD, CVPR, SIGIR, IJCAI) between 2015 and 2020 (46,774 papers). Ultimately, we found that 58.9% of those papers appeared in PWC. However these 58.9% of papers accounted for 89.3% of collective citations received by the 46,774 papers, suggesting that the missing papers are primarily lower impact papers.
> - There is recency bias stemming from the fact that PWC is relatively new and likely to focus on more recent papers. If we disaggregate the above sensitivity analysis by year, only 38.9% of top conference papers in 2015 are in PWC but 68.8% of 2020 papers are in PWC. That being said, citation coverage (as described above) in each year never drops below 86%.
> - The labeling of papers with datasets could independently have coverage bias, recency bias or spurious annotations that we cannot easily assess. We found that PWC labels papers with a dataset usage if the dataset name appears in the text of the paper (possibly in specific sections) even if the dataset is not actually used. To reduce these spurious labels, we only allowed usages to occur from papers labeled with the same tasks as the original dataset or dataset-introducing paper. We could probably gauge this bias with a small audit study.
> - Task labeling bias in PWC
> 	- There could also be coverage bias, recency bias or spurious annotations in the labeling of papers with tasks. An audit study here would require more expert discretion.
> 	- The strength and weakness of the PWC ontology of tasks and subtasks is that it is crowd-sourced. We’d like to think that this ontology represents some consensus on how tasks are related to each other, but there may be noise in the task annotation of small or new communities.
> 	- Because the PWC ontology is not a tree, it is hard to decide which level of task annotation to use when counting dataset transfers. To avoid double counting transfers at different levels of the ontology, we only focused on transfers between two tasks that were both parents of other subtasks. We also ran the analyses using all possible task-to-task transfers and it did not change our results.
> - Estimation issues
> 	- Gini, the adoption ratio, and the creation ratio are all unstable in small samples. To address this as best we could, we limited our analyses to tasks/task-years with at least 10 dataset usages and used a sample size-corrected Gini. This noise may be why we did not find any trends in the adoption ratio or creation ratio over time.
> 	- We chose Gini as our outcome for the inequality analyses because it is commonly used, but other scalar measures of discrete distributions produced similar results (Shannon Entropy based metrics, Simpson biodiversity index).

---

> > ### Author Response · Authors · 2021-09-30
> > **Response Continued**
> >
> > Weakness #2: **I believe some details not covered in the Appendix should be placed here, e.g., the calculated β estimates, more information on conducted AIC-based model selection, and a description of the reasoning that led you to choose linear or logistic regression in a particular scenario.**
> >
> > We chose to present mean-centered predictions instead of coefficients because coefficients are hard to interpret with many interactions. We included these interactions primarily because we thought the effects of each of our covariates might vary over time. We included the three-way interaction for task age, year, and task size because we thought that the dynamics of task communities of different sizes likely differ depending on how old the task is, but these dynamics may change as MLR has grown over time. Ultimately, the improvement in fit by adding this interaction is quite marginal (see fit statistics in Appendix). The exponentiated coefficients for the model are in the appendix, but we will make this more clear.
> >
> > We should clarify that we chose the presented model based on a combination of AICc and theoretical judgement. While there were similar models with an AICc within 2 of the fully restricted model (a rule of thumb), we thought all these covariates were appropriate to include in the model. We admit that we did not use other model selection/goodness-of-fit criteria (e.g., Bayesian Information Criterion) but have include these in the appendix in the interest of transparency.
> >
> > The rationale for modeling choices are in the paper, but were brief because of the word limit. We chose beta regression for the Gini coefficient because a) it is a standard choice for this outcome and b) the beta distribution is a very flexible distribution, particularly in our data where there are a lot of tasks close to 0 and 1 due to small sample sizes. Because of our sample sizes, we did have to smooth the Gini to avoid 0s. Other possible choices would be linear regression or logistic regression. Our data, even when transformed with standard transformations, violate standard homoscedasticity assumptions of linear regression. There is less theoretical or empirical motivation to use a logistic link function since our outcome is continuous, not a probability, and not binomially distributed.
> >
> > For the ratio analyses, we chose to formulate our outcomes as fractions of discrete events, so logistic regression where the outcome is binomially distributed is the most theoretically appropriate model.
> >
> > **Though parts of the process are debatable (e.g., due to insufficient description), I believe the analyses were conducted in a rather sound way and are correct.**
> >
> > We agree that this part of the paper should’ve been clearer by the deadline and have significantly rewritten the data section, describing in more detail how the dataset was created and cleaned. We have also provided a table with summary statistics to help clarify the data that were actually used after cleaning.
> >
> > **I noticed several minor problems:
> > you write that
> > p=0
> >  in #234, I believe it is a typo and there should be some non-zero value,
> > "e.g." in #46 lacks a comma despite the fact you add it in other parts of the paper,
> > footnote in #73 should be placed after the period,
> > "datset" instead of "dataset" in #169.**
> >
> > Thank you for noting these typos. The p-value in the aforementioned test is effectively 0 because it is so overpowered, but we have added significant digits.

---

### Official Review · Reviewer_ssmJ · 2021-09-22
**Excellent paper that reflects on the nature of datasets**

**Rating:** 10
**Confidence:** 5
**Correctness:** The data is what it is, and the analy…

**Strengths:**

The questions asked (and answered) in the paper are of high relevance to the field and to a track such as this. The methodology is sound and well suited to answer the questions. Additionally, the answers are bound to spark discussions and further work, as well as (hopefully) changes in the community. As such this paper is well worth publishing.

**Weaknesses:**

Per se the paper provides no new datasets, but it uses an existing one to great effect.

**Additional Feedback:**

Small points:
Line 73: Footnote 2 shoudl be after the period, not before
Line 158: the comma after e.g; it shoudl be e.g.,
Line 295: The parenthesis before Figure 4a should not have a leading space


**Clarity:**

The language is clear precise and to the point. No words are wasted and all points are well argued.

**Documentation:**

All data is available, although it is not strictly the contribution of the paper.

**Ethics:**

No ethical concern. Indeed this paper will help to spark ethical discussions and is thus "ethically good" in its own right.

**Relation To Prior Work:**

Since this paper does something very new relations to prior work are not that tight. However, it is well embedded in the existing literature.

**Summary And Contributions:**

This paper analyzed what and how datasets are used and finds a high concentration on a few datasets, usually from elite institutions. It holds a mirror to our community and raised  some very valid and pointed questions.

---

> ### Author Response · Authors · 2021-09-30
> **Response**
>
> We are both humbled and appreciative of your enthusiasm for the paper.  We have made the changes to the text recommended in the “Additional Feedback” section. While we agree that this isn’t a new dataset per se, we do pledge to make our data and code available to others.

---

### Author Response · Authors · 2021-09-30
**Thank you for your feedback!**

We thank the reviewers for the constructive feedback, for calling attention to areas of the paper that would benefit from more clarification/development, and for raising some interesting points for broader sociological discussion. Below we address comments by each of the reviewers individually.

---

### Decision · Program_Chairs · 2021-10-09

**Decision:**

Accept

**Comment:**

The reviewers found a lot of praise for this paper, and I strongly recommend acceptance. The findings are general, and I believe the material could make for an oral presentation that is interesting for a broad audience.